# Differentiated Thyroid Cancer in Children in the Last 20 Years: A Regional Study in Romania

**DOI:** 10.3390/jcm9113617

**Published:** 2020-11-10

**Authors:** Andreea-Ioana Ștefan, Andra Piciu, Simona Sorana Căinap, Katalin Gabora, Doina Piciu

**Affiliations:** 1Doctoral School of Iuliu Hațieganu, University of Medicine and Pharmacy, 400012 Cluj-Napoca, Romania; andreea.stefan24@gmail.com (A.-I.Ș.); gabora.katalin@yahoo.com (K.G.); doina.piciu@umfcluj.ro (D.P.); 2Department of Medical Oncology Iuliu Hațieganu, University of Medicine and Pharmacy, 400012 Cluj-Napoca, Romania; 3Department of Mother and Child, “Iuliu Hațieganu”, University of Medicine and Pharmacy, 400177 Cluj-Napoca, Romania; 4Department of Endocrine Tumors and Nuclear Medicine, “Prof. Dr. Ion Chiricuţă” Institute of Oncology, 400012 Cluj-Napoca, Romania

**Keywords:** pediatric differentiated thyroid cancer

## Abstract

Pediatric thyroid carcinoma is a current area of interest, because there are few studies in this field; the current classification and treatment guidelines are extrapolated from adults, sometimes leading to overly aggressive treatments or incomplete treatment of the disease. The purpose of this retrospective study is to analyze the presentation, diagnosis, treatment, complications, and outcome of children diagnosed with thyroid cancer in the last two decades (2000–2018) at the Oncological Institute “Prof. Dr. Ion Chiricuță” Cluj-Napoca (IOCN) Romania, a tertiary center in a region with environmental influences from both the nuclear fallout of the Chernobyl accident and from iodine deficiency. The results were compared with those obtained in a previous study carried out in the same institution between 1991–2010, and with those obtained in a similar study carried out in Netherlands between 1970–2013, a cohort of children not exposed to the post-Chernobyl fallout. We statistically analyzed 62 patients. Papillary form was present in the majority of cases, and we highlighted the occurrence of thyroid microcarcinoma in children. A total of 85.4% of patients received radioiodine, the total activity being significantly lower compared with the data from literature. In our study, the prognosis of the disease was excellent, with 100% overall survival.

## 1. Introduction

Pediatric thyroid carcinoma is a current area of interest, because there are few studies in this field; therefore, the current classification and treatment guidelines are extrapolated from adults, sometimes leading to overly aggressive treatments or incomplete treatment of the disease [1,2,3]. This pathology occurs relatively rarely in pediatric age, with an incidence at children of 0.4/100,000 (0–4 y) and 1.5/100,000 (15–19 y) [4].

The most frequent histopathological type of thyroid cancer (90% of all cases) is differentiated thyroid cancer (including papillary thyroid carcinoma (PTC) and follicular thyroid carcinoma (FTC) types), with a higher incidence in girls [5,6,7]. Although in children, the diagnosis is finally established in more advanced stages compared with adults, the disease prognostic is very good, with the overall survival at 30 years being over 95% [7,8,9,10,11,12].

The treatment is mostly based on adult protocols, but in 2015, the American Thyroid Association (ATA) published a recommended guideline of diagnostic and treatment for children with this pathology [5]. Although the prognostic of thyroid cancer in children is excellent, the post-therapeutic morbidity is high because of secondary hypoparathyroidism; recurrent, post-surgical laryngeal nerve injury; and salivary, lacrimal, and gonadal gland lesions (due to 131-I activity) [10,13,14].

There are a few studies in Europe concerning thyroid cancer in children. Almost all involve small cohorts, but as the number of cases continuously increases, more regional and national studies are needed for a better understanding of this pathology, and to create the diagnostic and therapeutic guidelines.

The purpose of this study is to analyze the clinical features, diagnosis protocol, treatment administered, any possible complications, and the post-therapeutical evolution of the children diagnosed with differentiated thyroid cancer (papillary and follicular type) in the period 2000–2018, at the Oncological Institute “Prof. Dr. Ion Chiricuță” Cluj-Napoca (IOCN), one of the national reference centers on oncological diseases for the northern, northwestern, and central areas of Romania (Figure 1). These regions in Romania are among the most affected areas by the Chernobyl accident in 1986 [15], the Romanian population being classified in 2002 by the World Health Organization as having an average iodine deficiency, with an insufficient intake of additional iodine [16]. For this reason, we also want to know if the 1986 nuclear accident still has an impact on this pathology at a considerable distance (14 to 24 years). Finally, we will compare the results with those obtained in a previous study carried out in the same institution between 1991–2010 [2], and with those obtained in the study carried out in Netherlands [4] between 1970–2013, a cohort of children not exposed to the post-Chernobyl fallout. In the section results, we will make a short presentation of the other identified types of thyroid cancer (medullary and anaplastic), but these cases are not included in the further statistical analysis.

## 2. Patients and Methods

### 2.1. Study Design and Population

This study is a retrospective cohort study, including children aged between 0 and 18.6 years who were diagnosed with thyroid cancer, from the database of the Oncological Institute “Prof. Dr. Ion Chiricuță”, Cluj-Napoca, between 1 January 2000 and 31 December 2018. 

After obtaining the approval of Ethical Committee of “Iuliu Hațieganu” University of Medicine and Pharmacy, Cluj-Napoca (number 58/11) and of “Prof. Dr. Ion Chiricuţă” Institute of Oncology, Cluj-Napoca (number 167/5), both in February 2020, for initiation of this study, the data collection was done retrospectively from the medical files of patients. All patients have previously signed, through their parents or legal tutors, the IOCN informed consent on participation in scientific studies.

### 2.2. Data Collection 

Data collection from the patient medical file included any past medical history, patient general information, the diagnosis, and the therapeutic protocol used. In the majority of cases, the surgery was performed in secondary centers, the surgical protocol was briefly noted in the transfer files, and the histopathological information was extracted from the original pathology report in the medical file. Because the tumor–node metastasis (TNM) classification of malignant tumors was changed several times within the period covered by this study, the tumor stage was (re)classified according to the seventh edition of the TNM classification, in order to unify the database [17,18]. Thus, the classification of the cases by age groups, according to the ATA recommendations, was possible [5]. We collected the information concerning 131-I administration (number of administrations, total dose activity), and the scintigraphy results, whole-body scans, or Positron Emission Tomography – Computed Tomography (PET–CT) from the patient medical file, including the nuclear medicine file. To calculate the cumulative 131-I activity administered, we consider all the 131-I administrations (ablative, therapeutic, and diagnostic). 

The biochemical results, including the level of thyroid-stimulating hormone (TSH), anti-thyroglobulin (anti-Tg), and thyroglobulin (Tg) were collected from the laboratory reports. In all the cases, the first 131-I administration was at 4–6 weeks after the surgery (with a maximum of 3 months). For all the patients, the 131-I administration was proceeded by thyroid-suppressing hormone therapy withdrawal, determining TSH stimulation.

In order to prevent the evolution of the disease, and to prevent recurrence, the TSH must be maintained at a suppressing serological level [4,5]. The median +/− standard deviation of the TSH values was 0.483 +/− 0.811 micro IU/mL, the majority of the cases having a light suppression of TSH.

### 2.3. Study Definitions

The date of diagnosis was defined as the date of the histopathological confirmation of thyroid carcinoma, and the follow-up time was the interval between the date of diagnosis until the date of the patient’s last known assessment or the date of the patient’s death. Age groups are considered depending on the age at diagnosis: group 1 (0–10 y), group 2 (11–14 y), and group 3 (15–18.6 y).

The Romanian pediatric population is defined as all the children under 18 years old. In this study, considering that thyroid cancer can be in an incipient phase much earlier than the date of diagnosis, we will take into consideration an upper age limit of 18.6 years at the time of diagnosis.

All conditions reported in the medical file were included in our study. Any forms of post-thyroidectomy induced hypoparathyroidism were reported. Transient hypoparathyroidism is defined as the need of calcium or vitamin D supplements for a duration of the treatment of less than six months, while permanent hypoparathyroidism has a duration of more than six months. Any injury of the recurrent laryngeal nerve by tumoral invasion or during surgery mentioned in the medical record was noted in the database.

Complete remission is defined as the absence of clinical, scintigraphy or radiological signs of the disease in association with an undetectable serum Tg after six months from the last administration on 131-I, under TSH stimulation.

Incomplete remission (biochemical persistence) is defined as the absence of the radiological or scintigraphy signs of the disease in the presence of positive tumoral markers (Tg, anti-Tg), and the persistent disease as the absence of remission. Recurrent disease is defined as the presence of thyroid cancer (radiological, scintigraphy, or serological) after remission.

Patients diagnosed with papillary thyroid cancer are classified, by ATA recommendation [5], according to the risk of recurrence: low risk (T1–T2, N0, M0), intermediate risk (any T3 or N1), or high risk (any T4 or M1), where T—tumor size, N—lymph nodes status, M—metastases present or absent. Besides the risk of recurrence, we have included in the database information on the immunochemistry analysis or genetic analysis (where applicable).

### 2.4. Statistical Analysis

The subject of the statistical analysis was differentiated thyroid cancer (papillary and follicular); any cases of medullary and anaplastic thyroid cancer were excluded from the study. We compared different types of groups using, for categorical variables, the Hi2 test or the Fisher test (if the conditions for the Hi2 test were not met), and for continuous variables without a normal distribution, the Mann–Whitney U and Kruskal–Wallis tests. Statistical analyzes were performed using IBM SPSS Statistics software version 22 (https://www.ibm.com/support/pages/spss-statistics-220-available-download, avaible for download from https://www.ibm.com/support/pages/downloading-ibm-spss-statistics-22). A value of *p* < 0.05 was considered statistically significant.

## 3. Results

### 3.1. Thyroid Cancer (Differentiated, Anaplastic, Medullary)

There were 71 patients included in the study (Figure 2, PRISMA diagram). Of these, due to technical reasons, three files could not be accessed, and one patient was investigated in a family context of medullary thyroid cancer, although the diagnosis was not confirmed. The following types of thyroid cancer were identified: papillary thyroid cancer = 56 cases, follicular thyroid cancer = 6 cases, medullary thyroid cancer = 4 cases, and anaplastic thyroid cancer = 1 case.

This study focused on the analysis of differentiated thyroid cancer, which includes the papillary and follicular type. The other types of cancer are not subject to this statistical analysis. Still, according to the National Health Insurance platform of 1 July 2020, only one patient was registered as deceased, namely the one diagnosed with anaplastic thyroid cancer, which is very aggressive and has a poor prognosis. Thus, in this cohort, the survival is 100% for differentiated and medullary thyroid cancer, and the mortality is 100% for anaplastic thyroid cancer.

#### 3.1.1. Differentiated Thyroid Cancer

After excluding the patients with non-differentiated histologies, 62 cases met the inclusion criteria. The distribution was the following: sex ratio (female/male (F/M)) was 4.1/1.0, namely 50 girls and 12 boys, and the average age of diagnosis was 13.1 years (8.2–18.0 years). Subdivided into age groups, the patients were distributed as follows: group 1–6 cases = 9.67%, F/M = 6/0; group 2–23 cases = 37%, F/M = 18/5; group 3–33 cases = 53.2%, F/M = 26/7. The average follow-up of these cases was 4.8 years (0.7–18.4 years).

#### 3.1.2. Baseline Characteristics

Table 1 provides the baseline characteristics. 

Regarding histology, 56 cases (90.4%) were classified as papillary thyroid cancer (PTC), and 6 cases as follicular thyroid cancer (FTC) (9.6%). The histological subtypes were papillary follicular variant, with 13 cases (20.9%), of which one developed on thyroglossal cyst; mixed (follicular and papillary), with 7 cases (11.2%); two cases (3.22%) of the cystic papillary variant, one of which developed on thyroglossal cyst; one case (1.61%) of the island variant; and one case (1.61%) classified as follicular and papillary with island areas. One case classified as PTC was described as the solid trabecular variant, and another case classified as FTC was described as the variant with oxyphilic cells. Tumor multifocality was recorded in 40.3% of cases. The size of the tumor focus varied from 0.15 cm to 6.4 cm, registering nine cases with tumor sizes of less than 1 cm, of which three had tumors less than 0.5 cm.

Regarding patient medical histories, in the database two brothers aged 13.2 and 14.4 years at the time of diagnosis are registered, both with PTC, without a history of malignancies in the family. Another patient with PTC also has a familial medical history of thyroid and testicular cancer (father). Among the past medical histories, two patients had relatives with other malignancies (in one case, a grandfather with pancreatic cancer, and in the second case, a father with nasopharyngeal carcinoma). 

Euthyroidism was present in 96.7% of cases; only two patients had hypothyroidism. Hashimoto’s thyroiditis was present as a comorbidity in five records (8%).

The clinical onset was determined by the deformation of the anterior cervical region in 19 cases (30.6%), of which eight cases (41.1%) were accompanied by dyspnea (swallowing disorders); thyroid ultrasounds revealed two cases with thyroid goiters (3.22%), one case with a thyroid nodule (1.61%), two cases of thyroid macro nodes (3.22%). In 15 cases (24.1%), the clinical onset was by palpation in the anterior cervical region of a thyroid nodule, or detection on a thyroid ultrasound of a macro node. In 10 cases (16.1%), the onset was by palpation of the cervical lymphadenopathy, which led to additional investigations. The triad of palpable cervical lymphadenopathy, deformity of the cervical region, and ultrasound-detected nodular goiter are present in a single case that also presents with hypothyroidism and Hashimoto’s thyroiditis.

Immunochemistry was performed in 15 patients (24.1%), of which there was one case of FTC (CK 19, CD 56, and HBME-1 positive).

In nine of the immunochemistry investigations (60%), the existence of CK 19 was determined, and it was positive in all the determinations performed. In five cases, the existence of CD 56 was tested, and it was positive in only one case. Galectin 3 was determined in five patients; of these, one result returned as inconclusive, two were positive, and two were negative. HBME-1 was determined in four cases; one result was inconclusive, and the other three were positives. Calcitonin was determined in four patients, and all returned negative. CEA-P and synaptophysin were determined in one case, and the result was negative. Chromogranin was determined in two cases; both results were negative. CK 7 was determined in one case, and the result was positive. Thyroglobulin (IHC) was positive in all seven samples analyzed. PANCK AE1/AE3 was tested in two cases; all tested positive. TTF1 tested positive only in the sample analyzed.

No genetic analysis were performed for the patients included in the study.

Cervical lymph node metastases were reported in 22 cases (35.4%), and pulmonary metastasis in six cases (9.6%). Two patients presented tumor infiltration in the parathyroid gland, and one of these was also a thymus infiltration.

Pathological changes and TNM staging were not different between the three age groups (Table 1).

### 3.2. Surgical Treatment

A total thyroidectomy was performed in 43.5% of cases, with reintervention in 41.9%, for a sum of total thyroidectomy in 85.4% of patients. The type of surgery during the first operative time differed depending on each case, as presented in Figure 3. Selective lymph node dissection was performed in 28.9% of cases.

Most of the patients underwent the surgical procedure in different centers; therefore, some surgical data might be missing, making statistical analysis impossible. 

The time between the two surgeries varied between 7 and 548 days (average 98 days). One case of PTC was excluded from the statistical analysis because of a 2687 day (seven-year) recurrence interval, the surgical procedure having been performed in 2001 in another university center, and in 2008 presenting recurrence of the underlying disease and undergoing another surgical procedure with a selective lymph node dissection.

#### 3.2.1. Surgical Complications

Recurrent laryngeal nerve injury occurred in three cases (4.8%), and hypoparathyroidism was observed in 10 patients (16.1%). There are statistical correlations between the advanced stage of the disease (T3T4) (*p* = 0.047), dissection of the cervical lymph nodes (*p* = 0.025), and recurrent laryngeal nerve damage as a postoperative complication (Table 2).

#### 3.2.2. 131-I Administration and TSH Suppression Therapy

Data regarding 131-I (radioactive – iodine) administrations is presented in Table 3. Fifty-three patients (85.4%) received 131-I therapy. The patient presenting recurrence of the disease after seven years was excluded from the analysis, because of the lack of documentation regarding any medical procedures undergone in the medical center where the initial surgery was performed. 

Only one male patient of 15.2 years diagnosed with PTC—mixed subtype, follicular and papillary, Stage I, T2N1bM0—received three courses of external radiotherapy immediately after surgery before being sent to the IOCN, although this procedure is not part of the official treatment protocol. 

### 3.3. Outcome

At the last known follow-up, 26 patients (41.9%) had a persistent disease, 16 of whom (61.5%) had a detectable Tg level (Table 4). One female patient presented a relapse of the disease seven years after the primary treatment of the PTC in a different medical center. 

The average activity of 131-I administrated was 186.68 mCi (6.9 GBq), with values between 17.88 and 990.15 mCi (0.66–36.6 GBq), Q1 = 70 mCi (2.59 GBq), and the average number of administrations being 2.38 with Q1 = 1 (Figure 4) [1,2,3,4,5,6,7,8,9]. We do not have records of the presence of pulmonary fibrosis, and no other neoplasms developed up to the time of publication of the data.

The status of the disease could not be evaluated for three patients, because they did not present at the scheduled reevaluation, and they were removed from the statistical analysis. 

The evolution of the disease was associated with lymph node involvement (N stage, *p* = 0.02), but did not associate with the T or M stages (*p* = 0.15 and *p* = 0.39, respectively). There were no differences between the three age groups or between the types of histology (Table 5). Also, there were no statistical differences in the evolution of patients depending on the type of tumor (multifocal or solitary), or in the initial size of the tumoral foci (Table 5).

The mean duration of follow-up was 8.03 years in those with the persistent disease, compared to 5.9 years in those with complete remission. Three patients in which the evolution of the disease cannot be assessed, because they did not present at the scheduled reevaluation, were removed from the analysis.

## 4. Discussion

For a better overview on this pathology, this regional study will be compared with the one carried out in the period 1991–2010 on a similar group (patients treated in IOCN) [2], as well as with a study carried out at the national level in the Netherlands, a country unaffected by the accident from Chernobyl [19], which analyzes pediatric patients in the period 1979–2013 [4].

The number of thyroid cancer cases registered in our study over a period of 19 years (*n* = 62) was similar to that in the study with a period of 20 years (*n* = 63), with a note that the appearance of new cases registered an increasing trend (Figure 5). In the Netherlands, over a period of 35 years, 170 patients were diagnosed, and 105 cases were included in the study.

The affected children in our study were born mostly after 1986 (date of the Chernobyl accident), with a large number of cases recorded among newborns in 1996, 1999, and 2000 (six patients) at 10, 13, and 14 years after the Chernobyl accident, respectively (Figure 6). It remains to be determined whether this upward trend in the appearance of thyroid cancer in children is due to residual radiation from the nuclear disaster that still affects the country’s population, or if there are other triggers to be identified.

The gender distribution is similar in all three studies, with the preponderance affecting female subjects; this data consistent with those in the literature.

The predominant histopathological variant was papillary thyroid cancer in all three studies. The presence of lymph node metastases was similar in the two regional studies conducted at IOCN, with 35.4% in 2000–2018 and 38.8% in the period 1991–2010, with a slight increase in cases in the national Dutch study, at 46%. Distant metastases were exclusively pulmonary in the Romanian subjects, with a recent increase, recorded in our study, up to 9.6% cases with lung metastases, compared to the previous study (1991–2010) in which there were only in 5.5% of cases with lung metastases. In the Dutch study, there were 13.3% cases with distant metastases, of which 91.6% were lung and 9.09% were bone metastasis.

The survival of thyroid cancer patients is excellent; this data was confirmed by the two other studies.

The surgical treatment was different in the two countries. In the Netherlands, all the patients had a total thyroidectomy. In Romania, in the period 1991–2010, all the children had total thyroidectomy except for one case, considered very low risk and being previously submitted to total lobectomy [2]. In the 2000–2018 period, the number of total thyroidectomies decreased (85.4%); in 51% of cases, the thyroidectomy was a single procedure. Lymph node dissection was performed in 43.8% of Dutch patients, similar to our study (43.5%).

In all studies, patients were surgically treated in different centers, no center being focused on pediatric thyroid pathology. Post-surgical complications, such as recurrent laryngeal nerve damage, was present in 20.9% of cases in Romania, compared to 12.35% in the Netherlands, and hypoparathyroidism in 16.1% of Romanian patients, compared to the Dutch study (38.95%). The appearance of post-surgical complications is correlated with the degree of staging (T3T4, *p* = 0.047), and with the cervical lymph node dissection (*p* = 0.025, see Table 2).

The median cumulative activity of 131-I was 186.68 mCi (6.9 GBq), with Q1 = 70 mCi (2.59 GBq), the average number of administrations being 2.38 [1,2,3,4,5,6,7,8,9], with Q1 = 1. These results are similar to those of the period 1991–2010. Given extreme values with single administrations, we will consider the Q1 of the 131-I dose (Figure 4). The mean 131-I cumulative activity is significantly lower than that recorded for the same population segment in the Netherlands (5.66 GBq).

Higher TNM stages, lymph node involvement, and distant metastases (T3–T4, N1a–N1b, and M1) were independently associated with higher total 131-I activity (*p* < 0.005). The number of 131-I administrations correlated with a higher T classification (T3T4, *p* = 0.005) and with lymph node invasion (*p* = 0.005), but there is no correlation with the presence of distant metastases (*p* = 0.059).

Cumulative 131-I activity and the number of 131-I administrations during initial and subsequent treatment did not differ by age groups (*p* = 0.066, *p* = 0.217) Table 3. There were no adverse effects of 131-I administration, such as pulmonary fibrosis or other malignancies, as described in the literature [3].

The 2000–2018 period study showed a complete remission in 51.6% of cases, and persistent disease in 41.9% of patients (61.5% of them with incomplete remission). In the period 1991–2010, 12.5% of patients were in incomplete remission. In the Dutch population, 8.6% of cases had persistent disease.

In our study, the persistent disease was correlated only with lymph node invasion (N1a–N1b), with *p* = 0.020, but not with the T classification, tumor type (multifocal or solitary character), tumor focus size, distant metastases, or age groups, as presented in Table 5.

The limitations of this study are that this is a retrospective study, all the data being collected from medical files, and some of which were incomplete. Almost all the patients were referred to IOCN only after their diagnosis. As the surgical treatment was performed in many different centers, the current study presents a heterogeneity of surgical procedures performed and pathology reports. 

Given the rarity of this pathology, we propose that all pediatric patients to be operated on in national centers, in order to standardize the surgical treatment and reduce postoperative complications.

It is imperative to have a national database that centralizes all cases, in order to be able to adjust the therapeutic protocols.

## 5. Conclusions

Differentiated thyroid cancer in children is a rare pathology, but with an upward trend, mostly due, in our country, to the Chernobyl nuclear disaster in 1986, the consequences of which are still visible. Despite that, other factors that could cause the apparition of thyroid cancer in children should be researched. Histopathological analysis of the study highlights the papillary and follicular types of thyroid cancer proportionally to those described in the literature. Small tumor foci (under 1 cm) in all cases of papillary thyroid carcinoma highlighted the presence of thyroid microcarcinoma in children. Low total 131-I activity in this age group was associated with good results, excellent prognosis of the disease, and 100% overall survival.

## Figures and Tables

**Figure 1 jcm-09-03617-f001:**
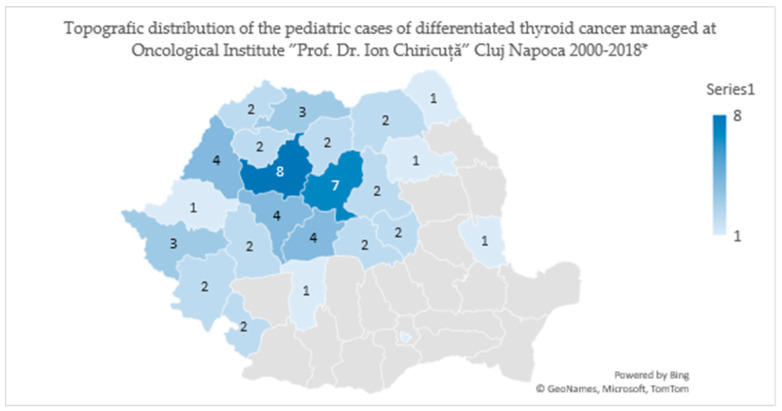
Topographic distribution of thyroid cancer cases in children treated at the Oncological Institute “Prof. Dr. Ion Chiricuță” Cluj-Napoca (IOCN) from 2000–2018. * The case from Bucharest comes, in fact, from the Republic of Moldova; the patient was diagnosed in a medical center in Chisinau. There the therapy was initiated, and later the family decided to move to Romania, with an official address in Bucharest, and went to IOCN to continue specific treatment.

**Figure 2 jcm-09-03617-f002:**
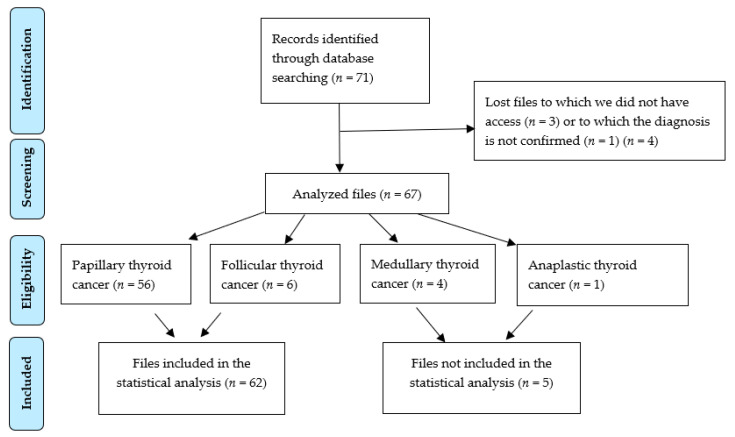
Patient selection criteria.

**Figure 3 jcm-09-03617-f003:**
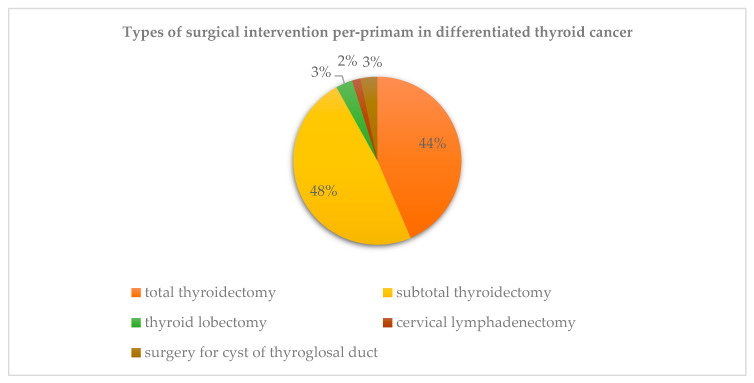
Types of the first surgical intervention in differentiated thyroid cancer.

**Figure 4 jcm-09-03617-f004:**
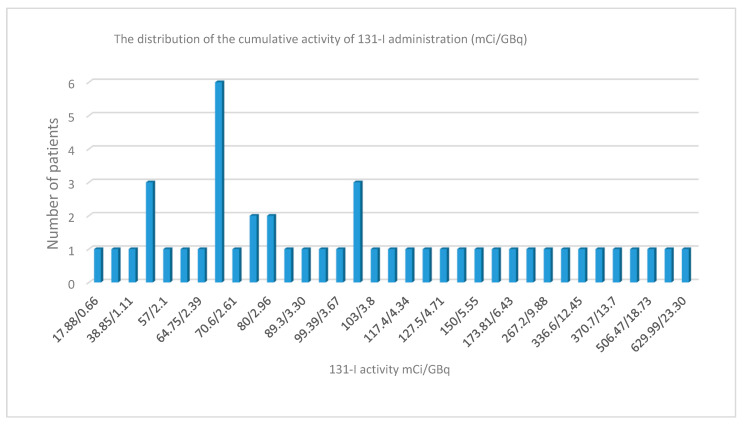
The distribution of the cumulative activity of 131-I administration (mCi/GBq).

**Figure 5 jcm-09-03617-f005:**
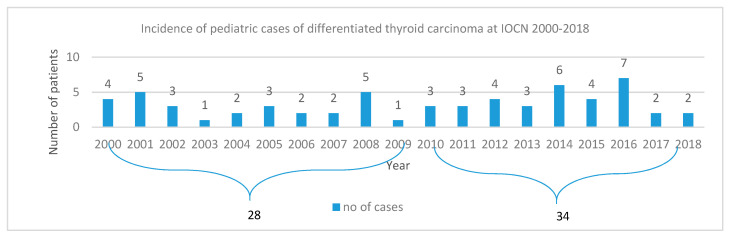
Incidence of pediatric differentiated thyroid carcinoma at “Prof. Dr. Ion Chiricuţă” Institute of Oncology (IOCN) in 2000–2018.

**Figure 6 jcm-09-03617-f006:**
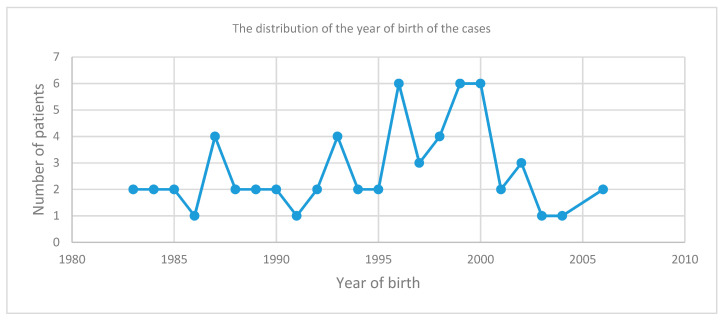
The distribution of the year of birth of the pediatric differentiated thyroid cancer cases in IOCN diagnosed in the period 2000–2018.

**Table 1 jcm-09-03617-t001:** Baseline characteristics.

Variable	All Patients (*n* = 62)	0–10 y (*n* = 6)	11–14 y (*n* = 23)	15–18 y (*n* = 33)	*p*-Value ^a^
**Sex, *n* (%)**					0.655
Male	12 (19.3)	0	5 (21.7)	7 (21.2)	
Female	50 (80.6)	6 (100)	18 (78.2)	26 (78.7)	
**Age at diagnosis, years**					n.a.
Median (range)	13.1 (8.2–18)	9.6 (8.2–10.8)	13.6 (11.7–14.9)	16.5 (15.1–18)	
**Histology, *n* (%)**					0.540
Papillary	56 (90.3)	5 (83.3)	22 (95.6)	29 (87.8)	
Follicular	6 (9.6)	1 (16.6)	1 (4.3)	4 (12.1)	
**Primary tumor size, cm**					0.587
Median (range)	2.15 (0.15–6.4)	1.9 (0.4–3)	2.1 (0.6–4.5)	2.14 (0.15–6.4)	
**Localization, *n* (%)**					0.422
Unilateral	45 (72.5)	4 (66.6)	14 (60.8)	27 (81.8)	
RTL	27 (43.5)	2 (33.3)	9 (39.1)	16 (48.4)	
LTL	15 (24.1)	2 (33.3)	4 (17.3)	9 (27.2)	
RTL + Isthmus	1 (1.6)	0	1 (4.34)	0	
LTL + Isthmus	2 (3.2)	0	0	2 (6.06)	
Bilateral	12 (19.3)	1 (16.6)	6 (26)	5 (15.1)	
Other ^b^					
Isthmus	1 (1.6)	0	1 (4.3)	0	
Thyroglossal duct	2 (3.2)	1 (16.6)	1 (4.3)	0	
Unknown	2 (3.2)	0	1 (4.3)	1 (3.03)	
**Multifocality, *n* (%)**					0.218
No	37 (59.6)	3 (50)	11 (47.8)	23 (69.6)	
Yes	25 (40.3)	3 (50)	12 (52.1)	10 (30.3)	
**TNM stage, *n* (%)**					0.856
**T**					
T1–T2	39 (62.9)	3 (50)	15 (65.2)	21 (63.6)	
T3–T4	23 (37)	3 (50)	8 (34.7)	12 (36.3)	
**N**					0.390
N0	26 (41.9)	1 (16.6)	8 (34.7)	17 (51.5)	
N1a–N1b	22 (35.4)	2 (33.3)	11 (47.8)	9 (27.2)	
N*x* ^b^	14 (22.5)	3 (50)	4 (17.3)	7 (21.2)	
**M**					0.448
M0	56 (90.3)	5 (83.3)	20 (86.9)	31 (93.9)	
M1 ^c^	6 (9.6)	1 (16.6)	3 (13)	2 (6)	
Lung	6	1	3	2	
Bone	0	0	0	0	
**Surgery, *n* (%)**					n.a.
Total thyroidectomy	27 + 26 = 53	2 + 4 = 6	10 + 10 = 20	15 + 12 = 27	
(S1 + S2)	(85.4)	(100)	(86.9)	(81.8)	
**Lymph node dissection**					0.183
None	40 (64.5)	3 (50)	15 (65.2)	22 (66.6)	
Central LND	3 (4.8)	2 (33.3)	1 (4.3)	0	
LND incl. Lateral levels	15 (24.1)	1 (16.6)	6 (26)	8 (24.2)	
Unknown	4 (6.4)	0	1 (4.34)	3 (9.09)	

Abbreviations: LND, lymph node dissection; n.a., not applicable; S1, first surgery; S2, second surgery; RTL, right thyroid lobe; LTL, left thyroid lobe. ^a^ Differences tested between the three age groups. Missing or unknown values were excluded from statistical testing. ^b^ The *x* indicates that there has been no assessment of that tumor characteristic, or information about that characteristic was not available. ^c^ Summarized as one variable for statistical testing.

**Table 2 jcm-09-03617-t002:** Surgical complications.

	Hypoparathyroidism, n (%)			Recurrent Laryngeal Nerve Injury, n (%)				
Group	Present *	n.a.	*p*-Value ^a^	Left	Right	Bilateral	n.a.	*p*-Value ^a^
All patients (*n* = 62)	10 (16.1)	52 (83.8)		1	1	1	59 (95.10)	0.047
T1–T2 (*n* = 39)	6 (15.3)	33 (84.6)	0.596	0	0	0	39 (100.00)	
T3–T4 (*n* = 23)	4 (17.3)	19 (82.6)		1 (4.34)	1 (4.34)	1 (4.34)	20 (8.69)	
No LND (*n* = 40)	5 (12.5)	35 (87.5)	0.102	0	0	0	40 (100.00)	0.025
LND (*n* = 18)	4 (22.2)	14 (77.7)		1 (5.55)	1 (5.55)	1 (5.55)	16 (88.80)	
LND unknown (*n* = 4)	1 (25.0)	3 (75.0)		0	0	1 (25.00)	3 (75.00)	

* We cannot assess from the information in the file if the hypoparathyroidism was transient or permanent, so we only evaluated the presence of hypoparathyroidism. Abbreviations: LND, lymph node dissection; n.a., not applicable. ^a^ Differences tested between T1–T2 and T3–T4, with no LND. Missing or unknown values were excluded from statistical testing.

**Table 3 jcm-09-03617-t003:** Administration of 131-I.

Group	Cumulative 131-I Activity		*p*-Value ^a^	131-I Therapeutic Administrations, *n*	*p*-Value ^a^
	mCi	GBq			
All patients (*n* = 52) ^b^	186.68 (17.88–990.15)	6.90 (0.66–36.63)		2.38 (1–9)	
T1–T2 (*n* = 29)	100.34 (17.88–351.10)	3.71 (0.66–12.99)	0.001	1.65 (1–5)	0.005
T3–T4 (*n* = 23)	295.54 (38.85–990.15)	10.9 (1.43–36.63)		3.30 (1–9)	
N0 (*n* = 19)	88.85 (17.88–267.20)	3.28 (0.66–9.88)	˂0.001	1.57 (1–5)	0.005
N1a–N1b (*n* = 19)	344.35 (38.85–990.15)	12.74 (1.43–36.63)		3.54 (1–9)	
N*x* (*n* = 14)	105.46 (50–265.51)	3.90 (1.85–9.82)		1.85 (1–3)	
M0 (*n* = 46)	153.88 (17.88–629.99)	5.69 (0.66–23.30)	0.022	2.15 (1–6)	0.059
M1 (*n* = 6)	438.14 (64.75–990.15)	16.21 (2.39–36.63)		4.16 (1–9)	
0–10 y (*n* = 6)	79.42 (30.0–200.7)	2.93 (1.11–7.42)	0.066	1.66 (1–3)	0.217
11–14 y (*n* = 21)	258.02 (17.88–990.15)	9.54 (0.66–36.63)		3.04 (1–9)	
15–18 y (*n* = 25)	152.35 (50–629.99)	5.63 (1.85–23.30)		2 (1–6)	

All data expressed as median (range). ^a^ Differences tested between T1**–**T2 and T3**–**T4, N0 and N1a**–**N1b, M0 and M1, and age groups. T*x*, N*x*, and M*x* were excluded from statistical testing. ^b^ Administered 131-I activity for one case was registered for the treatment of the recurrence; the initial treatment in the moment of diagnosis was not at the IOCN. Eight patients did not receive 131-I treatment; administered 131-I activity was unknown in one patient. Therefore, *n* = 52 instead of *n* = 62.

**Table 4 jcm-09-03617-t004:** Patients with persistent disease.

Sex and Age at Diagnosis, y	Follow-up, y	Histology, TNM *	Initial Treatment	Evidence of Disease	Localization of Disease
F, 15.1	2.1	FTC, T3aN0M0	TT, 131-I	Scintigraphy	Thyroid bed
F, 14.9	5.1	PTC, T3N1bM0	TT, LND, 131-I	PET–CT	Thyroid bed
F, 16.8	8.9	PTC, T4N1bM0	TT, LND, 131-I	WBS, PET–CT, histology	Thyroid bed, cervical lymph node
F, 14.0	3.4	PTC, T4aN1bM1	TT, LND, 131-I	WBS, PET-CT, CT	Cervical lymph node lung
F, 8.2	5	PTC, T2NxM0	TT, 131-I	WBS	Thyroid bed
F, 15.7	7.2	PTC, T3N1bM1	TT, 131-I	WBS	Cervical lymph node, thyroid bed, lung
M, 13.8	5.8	PTC, T3N1bM1	TT, LND, 131-I	WBS	Cervical lymph node, mediastinum, lung
F, 14.3	10.6	PTC, T4N1bM1	TT, LND, 131-I	WBS	Cervical lymph node, lung
F, 12.0	9.4	PTC, T3N1M0	TT, LND, 131-I	WBS	Cervical lymph node
F, 14.7	4.9	PTC, T3N1bM0	TT, 131-I	WBS	Thyroid bed
F, 15.9	2.4	PTC, T2N0M0	STT	Biochemical **	
F, 17.3	0.9	PTC, T1aN0M0	TT	Biochemical	
F, 8.5	4.4	PTC, T3N1bM0	TT, LND, 131-I	Biochemical	
M, 15.2	4.2	PTC, T1bN1bM0	TT, LND, 131-I	Biochemical	
F, 10.5	7.6	PTC, T1aNxM0	TT, 131-I	Biochemical	
F, 15.5	12.3	PTC, T2N0M0	TT, 131-I	Biochemical	
M, 12.2	11.8	PTC, T1N1aM0	TT, 131-I	Biochemical	
F, 14.0	10.1	PTC, T2N1M0	TT, LND, 131-I	Biochemical	
F, 12.0	14.9	PTC, T4N1M0	TT, LND, 131-I	Biochemical	
F, 12.8	10.7	PTC, T2aNxM0	TT, 131-I	Biochemical	
F, 16.0	7.8	PTC, T2NxM0	TT, 131-I	Biochemical	
F, 17.8	7.7	FTC, T2N0M0	TT, LND, 131-I	Biochemical	
F, 13.3	18.4	PTC, T2N0M0	TT, 131-I	Biochemical	
F, 18.0	17.4	PTC, T2N0M0	TT, LND, 131-I	Biochemical	
M, 15.2	8.8	PTC, T1N0M0	STT, 131-I	Biochemical	
M, 14.8	7.1	PTC, T3N1bM0	TT, LND, 131-I	Biochemical	

* Initial TNM classification; ** Tg, anti-Tg. Abbreviations: TT, total thyroidectomy; STT, subtotal thyroidectomy.

**Table 5 jcm-09-03617-t005:** Outcome.

Group	Remission *n* (%)	Recurrence *n* (%) **	Persistent Disease *n* (%)	*p*-Value	Unknown *n* (%)
All patients (*n* = 62)	32 (51.6)	1 (1.6)	26 (41.9)		3 (4.8)
T1–T2 (*n* = 39)	23 (58.9)		14 (35.8)	0.155	2 (5.1)
T3–T4 (*n* = 23)	9 (39.1)		12 (52.1)		1 (4.3)
Multifocality *Yes (*n* = 22)	11 (50.0)		11 (50)	0.598	0
No (*n* = 35)	20 (57.1)		15 (42.9)		3 (100.0)
Tumoral foci (cm) * ˂1 cm (*n* = 9)	5 (55.5)		3 (33.3)	0.092	1 (11.1)
1–2 cm (*n* = 15)	13 (86.7)		2 (13.3)		
2–3 cm (*n* = 13)	6 (46.1)		6 (46.1)		1 (7.6)
3–4 cm (*n* = 8)	2 (28.6)		5 (71.4)		1 (12.5)
4–5 cm (*n* = 3)	1 (33.3)		2 (66.7)		
5–6 cm (*n* = 1)	0		0		1 (100.0)
6–7 cm (*n* = 1)	1 (100.0)		0		
N0 (*n* = 26)	17 (65.3)		8 (30.7)	0.020	1 (3.8)
N1a–N1b (*n* = 22)	6 (27.2)		14 (63.6)		1 (4.5)
N*x* (*n* = 14)	9 (64.2)		4 (28.5)		1 (7.14)
M1 (*n* = 6)	2 (33.3)		4 (66.6)	0.393	0
M0 (*n* = 56)	30 (53.5)		22 (39.2)		3 (5.3)
FTC (*n* = 6)	4 (66.6)		2 (33.3)	0.55	0
PTC (*n* = 56)	28 (50.0)		24 (42.8)		3 (5.35)
Age groups: 0–10 y (*n* = 6)	3 (50.0)		3 (50.0)	0.425	0
11–14 y (*n* = 23)	10 (45.5)		12 (54.5)		1 (4.3)
15–18 y (*n* = 33)	19 (63.3)		11 (36.7)		2 (6.06)

* Patients for whom there is no data on multifocality (one patient) or the exact size of the tumor focus (12 patients) were removed from the study. ** Only one patient had a recurrence of the disease after seven years from the initial diagnosis, established in another medical center with no medical record available, and it was not included in the subsequent statistical analysis.

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
