# Peer review of "Differentiated Thyroid Cancer in Children in the Last 20 Years: A Regional Study in Romania"

_jcm, 2020, doi:10.3390/jcm9113617_

Round 1
Reviewer 1 Report
The manuscript entitled ”Differentiated thyroid cancer in children in the last 20 years, a regional study in Romania” has useful information for readers who are interested in this field. However, there are some problems to be solved before publication.
- Line 39 The author mentioned “in more advanced stages”, however, all of children case are Stage I or II.
- Line 44 “do to secondary hypothyroidism,” → “due to secondary hypothyroidism,”
- Line 45 “do to 131-I activity” → “due to 131-I activity”
- Figure 2 is not correct. There were 71 “Records identified through database searching”. Of them, 4 cases were excluded and finally 67 cases were analyzed. The figure should show this procedure, however, seems to show 71+4=67.
- Table 1 The authors should indicate what are abbreviated to “LTD” and “LTS”.
- Line 220-221 “the reason for what it was not possible to analyze this information correctly. The authors should check this sentence grammatically.
- Table 2 On Line 1, “p value 0.102“ of “All patients” should be deleted. The word “Unknown” doesn’t seem to be correct in this matter.
- Table 5 The numbers of cases of “Tumoral foci” are not consistent with the total number.
Author Response
Review report 1
Distinguished reviewer thank you very much for your support and efforts in evaluating our manuscript.
Here are our responses point-by-point
The manuscript entitled ”Differentiated thyroid cancer in children in the last 20 years, a regional study in Romania” has useful information for readers who are interested in this field. However, there are some problems to be solved before publication.
- Line 39 The author mentioned “in more advanced stages”, however, all of children case are Stage I or II.
In differentiated thyroid cancer the actual AJCC 7thed staging splits the patients into two different groups according to age: under 55 years old and above 55 years old. All patients below 55 years-old (as are the children) are in stage I if they do not have distant metastases and stage II if distant metastases are present. Actually, children, no matter how advanced is the disease are in stage I or II. Higher stages do not exist in this age group.
- Line 44 “do to secondary hypothyroidism,” → “due to secondary hypothyroidism,”
Corrected accordingly
- Line 45 “do to 131-I activity” → “due to 131-I activity”
Corrected accordingly
- Figure 2 is not correct. There were 71 “Records identified through database searching”. Of them, 4 cases were excluded and finally 67 cases were analyzed. The figure should show this procedure, however, seems to show 71+4=67.
Thank you, we have corrected.
- Table 1 The authors should indicate what are abbreviated to “LTD” and “LTS”.
We have inserted in the legend
- Line 220-221 “the reason for what it was not possible to analyze this information correctly. The authors should check this sentence grammatically.
We have changed it.
- Table 2 On Line 1, “p value 0.102“ of “All patients” should be deleted. The word “Unknown” doesn’t seem to be correct in this matter.
Thank you, we have corrected.
- Table 5 The numbers of cases of “Tumoral foci” are not consistent with the total number.
Thank you, we have corrected.

Reviewer 2 Report
In their study, Stefan and Co-workers evaluated, through a retrospective study, a group of children (Group A, n = 62) with differentiated thyroid cancer in the frame time 2000-2018 in a tertiary centre in Romania.
The manuscript is very difficult to read, given the numerous lexical and grammatical errors.
The aims of the study are scattered along the manuscript:
- Comparison of the Group A with of a group of children (Group B) with diagnosis of thyroid cancer in the frame time 1991-2010 (Section Abstract)
- Presentation, Diagnosis, Treatment, Complications, Evolution (Section Introduction, “Purpose of the study”)
- Impact of the nuclear accident in Chernobyl (year 1986), Section Patients and Methods.
The section Results does not follow the outline of the “Purpose of the study. Besides, the Section starts with “Thyroid cancer (differentiated, anaplastic, medullary)”. The last two diseases are not included in the Purpose of the study. In the same Section, the part related to the survival of patients with medullary and anaplastic seems then not appropriate.
In several cases, Results are duplicated in the Discussion (entire identical sentences): see for example page 9, lines 254-256 and page 13, lines 334-336 (except from the lower limit of the range –7.88 vs. 17.88-, which is therefore wrong in one of the sentence).
It is unclear whether the sentence “papillary thyroid cancer” (page 3, line 120) refers to “differentiated thyroid cancer”
The abbreviations PTC and FTC (page 5, lines 162 and 163) are belatedly introduced in the text.
The introduction of the abbreviation TT and STT (page 11 line 271) is misplaced.
Author Response
Review report 2
Distinguish reviewer we appreciate the time and effort that you dedicated to providing feedback on our manuscript and are grateful for the insightful comments on and valuable improvements to our paper. We have incorporated the suggestions made by the reviewers. Those changes are highlighted within the manuscript.
Please see below, a point-by-point response to the reviewers’ comments and concerns. All page numbers refer to the revised manuscript file with tracked changes.
"In their study, Stefan and Co-workers evaluated, through a retrospective study, a group of children (Group A, n = 62) with differentiated thyroid cancer in the frame time 2000-2018 in a tertiary centre in Romania.
The manuscript is very difficult to read, given the numerous lexical and grammatical errors".
We have revised it for English language.
The aims of the study are scattered along the manuscript:
- Comparison of the Group A with of a group of children (Group B) with diagnosis of thyroid cancer in the frame time 1991-2010 (Section Abstract)
- Presentation, Diagnosis, Treatment, Complications, Evolution (Section Introduction, “Purpose of the study”)
- Impact of the nuclear accident in Chernobyl (year 1986), Section Patients and Methods.
We modified it; restructuring the objectives
The section Results does not follow the outline of the “Purpose of the study. Besides, the Section starts with “Thyroid cancer (differentiated, anaplastic, medullary)”. The last two diseases are not included in the Purpose of the study. In the same Section, the part related to the survival of patients with medullary and anaplastic seems then not appropriate.
Thank you for your suggestion. We have changed and insert necessary comments.
In several cases, Results are duplicated in the Discussion (entire identical sentences): see for example page 9, lines 254-256 and page 13, lines 334-336 (except from the lower limit of the range –7.88 vs. 17.88-, which is therefore wrong in one of the sentence).
We have changed it; rephrase it and delete the redundant information.
It is unclear whether the sentence “papillary thyroid cancer” (page 3, line 120) refers to “differentiated thyroid cancer”
Papillary thyroid cancer is a differentiated thyroid cancer ; we have introduced a comment in text.
The abbreviations PTC and FTC (page 5, lines 162 and 163) are belatedly introduced in the text.
We have changed it.
The introduction of the abbreviation TT and STT (page 11 line 271) is misplaced.
We have changed it accordingly

Round 2
Reviewer 2 Report
The manuscript has undergone a considerable process of revision, and appears in my opinion markedly improved. Good luck!